Spatiotemporal changes of eutrophication and heavy metal pollution in the inflow river system of Baiyangdian after the establishment of Xiongan New Area

Wang Yibing 1 2
Wang Yang 3
Zhang Wenjie 1
Yao Xu 1
Wang Bo 1
Wang Zheng wzhwangzheng@126.com 1 2
1 College of Forestry, Hebei Agricultural University , Baoding , China
2 Hebei Urban Forest Health Technology Innovation Center , Baoding , China
3 College of Land and Resources, Hebei Agricultural University , Baoding , China
Wang Jingzhe
Electronic publication date: 2022 May 3
Publication date: 2022
Volume: 10
Electronic Location ID: e13400
Received 2022 Jan 10; Accepted 2022 Apr 17
Copyright: ©2022 Wang et al.
Copyright year: 2022
Copyright holder: Wang et al.
License: This is an open access article distributed under the terms of the Creative Commons Attribution License, which permits unrestricted use, distribution, reproduction and adaptation in any medium and for any purpose provided that it is properly attributed. For attribution, the original author(s), title, publication source (PeerJ) and either DOI or URL of the article must be cited.
License URL: https://creativecommons.org/licenses/by/4.0/

Keywords: Spatiotemporal variation, Inflow river, Inland waters, Baiyangdian, Fuhe river, Eutrophic pollutant, Trace element, Pollution hotspot

Funding: The University Scientific and Technological Research Program in Hebei Province QN2020239 The Program of Modern Forestry Disciplinary Group of Hebei Agricultural University XK1008601519 The Returned Overseas Talent Program in Hebei Province C20190181 The Young Talent Program of Hebei Agricultural University 2D201732 This work was supported by the University Scientific and Technological Research Program in Hebei Province (QN2020239), the Program of Modern Forestry Disciplinary Group of Hebei Agricultural University (XK1008601519), the Returned Overseas Talent Program in Hebei Province (C20190181) and the Young Talent Program of Hebei Agricultural University (2D201732). The funders had no role in study design, data collection and analysis, decision to publish, or preparation of the manuscript.

==============================
Pollution in inflow rivers seriously endangers the water environment in downstream lakes. In this study, an inflow river system of the Baiyangdian–Fuhe river system (FRS) was investigated to display timely pollution patterns of eutrophication and heavy metals after the establishment of Xiongan New Area, aiming to reveal the weak parts in current pollution treatments and guide the further water quality management. The results showed that the pollution of eutrophication was worse than the heavy metals in FRS, with serious eutrophic parameters of ammonia nitrogen (NH4+-N) and chemical oxygen demand (COD). There were greatly spatiotemporal variations of the pollution in FRS. (1) Concentrations of NH4+-N and total phosphorus were all higher in summer and autumn, whereas, COD contents were higher in spring; the water quality index (WQI) of eutrophication linearly increased along FRS in summer and autumn, with pollution hotspots around the estuary area. (2) The pollution levels of plumbum exceeded cadmium (Cd) and chromium (Cr) but without strongly spatiotemporal changes; however, Cd and Cr in the town area and Cd in spring showed higher concentrations; the WQI of heavy metals showed single peak curves along FRS, with significantly higher values around the town area. Additionally, the four potential pollution sources: domestic sewage, traffic pollution, agricultural wastewater and polluted sediments were identified based on the pollution patterns and pollutant associations. These findings demonstrated current treatments failed to eliminate the pollution in some hotspots and periods, and the in-depth understanding of the pollution spatiotemporal patterns in this study, especially the pollution hotspots, serious periods and potential sources, are crucial to furtherly develop spatiotemporally flexible pollution treatment strategies.

Introduction

Water quality of inland waters is increasingly disconcerting for a long period due to the hazardous impacts of water deterioration in the world (Huang et al., 2019; Tong et al., 2017; Wang et al., 2021a). The water quality of inland waters is determined by numerous factors such as climate, hydrologic conditions, and anthropogenic activities (Sinha, Michalak & Balaji, 2017; Wang et al., 2021a; Zhuang et al., 2019). Generally, land use intensification and urbanization with increased population density have been considered as the most important driving factors for the declining water quality in inland waters (Huang et al., 2019; Tong et al., 2017; Zhou et al., 2017), which resulted in the increased wastewater discharge from households, agriculture and industry (Meng, Zhang & Shan, 2021; Tong et al., 2017; Zhou et al., 2017). In China, inland waters are generally surrounded by densely populated areas, and the severe contamination has occurred in these inland waters, such as Taihu, Dianchi and Poyang lakes (Lv et al., 2020; Wang et al., 2021a; Wu et al., 2017), as well as Baiyangdian (Li, Chen & Sun, 2021a; Meng, Zhang & Shan, 2021; Zhang et al., 2018). The water quality of inland waters has been extensively studied with special emphasis on topics, such as eutrophication (Sinha, Michalak & Balaji, 2017; Tong et al., 2017) and nutrient loading (Ni, Wang & Wang, 2016; Tao et al., 2020; Zhu et al., 2019). Eutrophication due to phosphorus and nitrogen pollution has posed a risk to the health and stability of the aquatic ecosystem, and ammonia nitrogen (NH4+-N), total phosphorus (TP) and chemical oxygen demand (COD) are three important parameters of eutrophication in inland waters (Li et al., 2021b; Meng, Zhang & Shan, 2021; Ni, Wang & Wang, 2016; Tong et al., 2017; Zhao et al., 2020). Water pollution caused by heavy metals also has caused widespread concern due to their health threat to aquatic biota and humans (Rajeshkumar & Li, 2018; Zhang et al., 2017; Zhang, Wu & Giesy, 2019). At present, heavy metal pollution displayed an increasing trend in many lakes and reservoirs in China (Huang et al., 2019), and a large number of heavy metals were still released from historical polluted sediments, causing a high risk of contamination to the aquatic ecosystem (Ke et al., 2017; Vu et al., 2017; Zhang et al., 2018).

Pollution in inflow rivers seriously endangers the water environment in downstream lakes. Comprehensive measures have been made regarding pollution reduction in inland waters in last decades (Chen et al., 2021a; Wang et al., 2021b; Zhou et al., 2017); however, the serious pollutions were still frequently observable due to the measures not taking serious consideration of the water pollution in inflow rivers (Gao et al., 2021; Lv et al., 2020; Wang et al., 2021a). The inflow rivers and the stream networks could collect domestic, agricultural and industrial wastewater from densely populated areas in the whole watershed, which have become the main reason for the water deteriorating in the downstream waters (Aubriot et al., 2020; Gao et al., 2021; Lv et al., 2020; Wang et al., 2021a; Wang et al., 2021b). In addition, the dominant contributors to the pollutants were presently changed from point source pollution (e.g., local industries) to the diffuse source pollution and internal pollution loading, which had greatly spatiotemporal variations along the inflow river systems (Huang et al., 2018; Wang et al., 2017; Xia et al., 2018; Zha et al., 2018; Zhu et al., 2019). A single pollution control method could not deal with the spatiotemporally varied pollution (Huang et al., 2018; He et al., 2020). Thus, water quality management strategies could hardly be successful unless the spatiotemporal variation of water pollution are taken into serious consideration, and spatiotemporal flexible measures should be furtherly made along the inflow rivers for the greater improvement of water quality (Aubriot et al., 2020; Gao et al., 2021; He et al., 2020; Wang et al., 2021a).

Water quality of inland waters in the North China Plain was found to be the poorest in China due to the high coverage of developed land (cities and cropland) and population density (Zhou et al., 2017). In addition, the relatively increased evaporation in this region created more protracted periods of drought in spring and early summer, which prolonged the water retention time and amplifying the deterioration of water quality in inland waters (Piao et al., 2010; Sinha, Michalak & Balaji, 2017). Baiyangdian is the largest lake in North China Plain, playing an irreplaceable role in maintaining the environmental health in this region (Cheng, Bo & Sun, 2018; Guo, Huo & Ding, 2015; Yi, Lin & Tang, 2020; Zhao et al., 2021). However, with the rapid development of industry and agriculture, tons of pollutants were discharged into the water of inflow river systems and Baiyangdian since 1980s, which not only severely contaminated of the water body but also the sediments and aquatic biota (Guo, Huo & Ding, 2015; Meng, Zhang & Shan, 2021; Tao et al., 2020). The Xiongan New Area started to be built around Baiyangdian in 2017, which aimed to establish an advanced new area for the coordinated development of ecology and economy in China (Xia & Zhang, 2017). Since then, the water quality of Baiyangdian has been greatly concerned (Cheng, Bo & Sun, 2018; Meng, Zhang & Shan, 2021; Zhao et al., 2021). The Chinese central and local governments have made substantial investments to environmental remediation in order to improve the water quality in Baiyangdian watershed to guarantee the healthy development of Xiongan New Area (Chen et al., 2021a; Xia & Zhang, 2017; Zhao et al., 2021). Presently, point source pollution has been basically controlled (Chen et al., 2021a; Li, Chen & Sun, 2021a). However, the water pollution in Baiyangdian was still in a high level, especially the severely eutrophic parameters of NH4+-N, TP and COD, as well as the serious heavy metal pollution of plumbum (Pb), cadmium (Cd) and chromium (Cr) in the water and sediments (Li et al., 2021b; Gao et al., 2019; Meng, Zhang & Shan, 2021; Zhao et al., 2021). The diffuse source pollution in this watershed became a main problem, such as agricultural sewage, domestic garbage and sediment release (Li et al., 2017; Li, Chen & Sun, 2021a; Tao et al., 2020; Zhu et al., 2019), which were carried by the inflow rivers from the whole watershed to contaminate the water environment in Baiyangdian and Xiongan New Area (Meng, Zhang & Shan, 2021; Tao et al., 2020; Zhao et al., 2020). Thus, eliminating the diffuse source pollution of the inflow rivers is critically important for the improvement of the water quality in Baiyangdian (Li, Chen & Sun, 2021a; Zhao et al., 2020; Zhao et al., 2021), and a flexible set of measures for water pollution control should be adopted considering the spatiotemporal differences of the diffuse source pollution (He et al., 2020; Huang et al., 2018; Tong et al., 2017). Surprisingly, our literature review has found that the spatiotemporal distributions and variability of water pollution in the inflow rivers of Baiyangdian have not been explored since the establishment of Xiongan New Area; in particular, we lack the timely understanding about the pollution hotspots, serious periods and potential sources in a whole inflow river system to distinguish the weak parts in current pollution treatments.

In this study, a typical inflow river system of the Baiyangdian–Fuhe river system (FRS) was selected, which is one of the main water sources of Baiyangdian and passes through a big city, as well as many towns and villages (Cheng, Bo & Sun, 2018; Guo, Huo & Ding, 2015). Thus, FRS received strong impact of anthropogenic activities and was considered as one of the most seriously polluted inflow rivers of Baiyangdian (Li et al., 2017; Liang et al., 2017). To better guide the pollution control strategies, we measured the water eutrophication (NH4+-N, TP and COD) and heavy metal pollution (Pb, Cd and Cr) of the FRS from the headstream to the estuary in three seasons to investigate the spatiotemporal pollution pattern, identify the pollution hotspots and periods, and make a timely assessment of the water quality after the establishment of Xiongan New Area. We aimed to reveal the weak parts in current pollution treatments, and provide scientific basis to make spatiotemporally flexible measures for water quality improvement in Baiyangdian watershed. This study could be a reference for the water pollution treatment and ecological restoration in other inland waters.

Materials & Methods

Study area

Baiyangdian Lake, located in north latitude 38°43′–39°02′, east longitude 115°38′–116°07′, lies in the semiarid warm temperate continental monsoon climate zone with four distinct seasons. The average annual precipitation is 539.7 mm, and 80% of the precipitation is concentrated in June to August (Cheng, Bo & Sun, 2018; Yi, Lin & Tang, 2020). Baiyangdian stays in the middle reaches of Daqing River System in Haihe River Basin, undertaking the floodwater storage of nine rivers, with a total area of 366 km2 and an average annual water storage capacity of 1.32 billion m3 (Xia & Zhang, 2017; Zhao et al., 2021). Baiyangdian is the biggest natural wetland in the North China Plain and locates in the south part of the Xiongan New Area, which has direct and significant influences on the ecological health of this region (Cheng, Bo & Sun, 2018; Xia & Zhang, 2017; Zhang et al., 2018). However, due to the influence of anthropogenic activities in recent decades, a large number of point and diffuse source pollutants flowed into Baiyangdian Lake through the upstream river systems, endangering the water ecological environment and causing serious eutrophication and heavy metal pollution (Guo, Huo & Ding, 2015; Li, Chen & Sun, 2021a; Meng, Zhang & Shan, 2021; Zhao et al., 2020).

Sample collection

A typical inflow river system of Baiyangdian–Fuhe river system (FRS) was selected to study the spatiotemporal pollution patterns and assess the water quality. Sampling was carried out sequentially along FRS from the headstream to the estuary area, taking into account of the topographic distribution, flow path distance and surrounding land uses, and a total of fourteen sampling sites were set up (Fig. 1). The source of FRS is located in the Mountain area in the west of Baoding. The area of the “headstream” in Fig. 1 is located near the mountain area and before the densely populated areas, thus we considered it is a part of the headstream of FRS. The water quality in FRS would not change too much during the water transformation in the mountain area due to the low anthropogenic activities and wastewater discharge. Therefore, we started our sampling from the “headstream” area in Fig. 1. River water was sampled in three seasons: spring (May, headstream was not collected), summer (July) and autumn (October) of 2020, respectively. In order to reduce experimental errors, at least four sample points were randomly selected in different areas in each sample site, and water samples at each sample point were taken at two depths: ∼0.1 m below the water surface (surface samples) and ∼0.3 m below the water surface (deep samples). Each sample was collected 150 ml water into a brown polyethylene bottle, which was acid cleaned and rinsed with surface water before sampling. The water samples were stored in a cooler with ice bags and then placed in a refrigerator at 4 °C after returning to our laboratory.

Figure 1 Distribution of sampling sites in one of inflow river systems of Baiyangdian–Fuhe river system (FRS).

Sampling sites in FRS were classified into five areas according to their different positions and surrounding land-use conditions along the sampling route: Headstream (F1-3), City (F4-8), Towns (F9-10), Farmland (F11-13) and Estuary (F14).

Pollutant concentration measurement

When water samples were transported to our laboratory, they were filtered using GF/F filters (Whatman, Kent Great Britain). Then each water sample was separated into 100 ml and 50 ml two parts. The 100 ml water sample was immediately used to determine concentrations of COD, NH4+-N and TP by the potassium dichromate method, nesslerization spectrophotometry and Mo-Sb Anti spectrophotometric method, respectively, according to the procedures of surface water quality measurements (HJ 828-2017, HJ 535-2009 and GB 11893-89) in China. The other 50 ml water sample was stored with 1.5 ml 68% HNO3 at 4 °C for the analysis of heavy metal in two weeks. Pb, Cd and Cr were determined using Atomic Absorption Spectrometry (AAS ZEEnit-700P). The precision of the instrument was checked through the chemical standards (Merck, Germany) with control blanks yielding a quantitative value of 100 ± 4.1% (Rajeshkumar & Li, 2018). Five-point calibration curves were used for the concentration measurement, and R2 values of calibration curves greater than 0.99 were accepted. Two replicates were measured for each sample, and the heavy metal concentrations in the blanks were subtracted from the sample values (Xia et al., 2018).

Water pollution evaluation

Considering the applicability of the evaluation methods, the water quality index (WQI) method (Gao et al., 2019; Wang et al., 2017) was used to analyze the comprehensive pollution conditions of FRS, and the water quality conditions were classified as Table S1. WQI of eutrophication (WQIE) was calculated according to Class III water standard (GB3838-2002) due to it being the present water quality requirement of Baiyangdian; and most of heavy metal pollutants were lower than Grade III water standard and needed to meet Grade I water standard as a natural reserve in future (Table S2), WQI of heavy metals (WQIHM) was calculated according to these two water standards in this study.

WQI was computed as follows:

(1) Ai=CiCsi

(2) WQI=1n∑i=1nAi

Ai—Pollution index of a certain pollutant (i);

Ci—Measured concentration of a certain pollutant (i);

Csi—Water quality standard of a certain pollutant;

n—Number of elements.

Spatiotemporal variation analysis

The remote sensing images of Fuhe River watershed in 2019 growing season were obtained from Landsat (http://www.gscloud.cn), and then ENVI Classic was used to classify the land use types. Based on the characteristics of surface feature spectrum and remote sensing image, as well as the distribution characteristics of the research object, we established training samples for supervised classification and visual interpretation of land use types with the reference of the national land use/cover classification system. We continuously optimized the classification results to ensure the accuracy of the data. After the data of pollutant concentrations in three seasons were combined with the GPS positioning of each sampling site, Arc-GIS was used to analyze the spatial and temporal changes of eutrophication and heavy metal pollution in FRS.

Data analysis

One-way analysis of variance (ANOVA) was conducted to compare the differences of the pollution parameters in FRS (least-significance difference, LSD), using SPSS 16.0 for Windows (SPSS Inc., Chicago, IL, USA, 2002). We checked the normality and homogeneity of variances for the ANOVAs using Shapiro–Wilk and Levene tests. Data were transformed to meet the assumptions of normality and homogeneity of variance where necessary. The significance of the differences among the median values of sampling areas were tested by Kruskal–Wallis one-way analysis. Pearson’s correlation analysis was used to perform correlations between the eutrophic parameters and heavy metals (Chen et al., 2021b; Guo et al., 2020). Principal component analysis (PCA) could explore the possible sources of heavy metals by reducing the dimensionality of the multivariate water pollutant dataset to 2–3 principle influencing factors, which commonly occurs in hydrochemistry (Guo et al., 2021; Ismail et al., 2016; Zhuang et al., 2019). In this study, Pearson’s correlation analysis and PCA were employed to identify potential sources and hotspots of heavy metal pollution in FRS. The average values of surface samples and deep samples in each sample site were used for the Pearson’s correlation analysis and PCA. All the Pearson’s correlation and PCA analyses were performed in the R platform (R Core Team, 2018). The R package of “FactoMineR” (Lê, Josse & Husson, 2008) was used to calculate the principle components, and the “factoextra” package (Alboukadel & Fabian, 2017) was used to extract and visualize the results.

Results

Extreme pollution in individual sampling site of FRS in three seasons

The changes of eutrophication and heavy metal pollution along the sampling route in FRS are showed in Fig. 2. Three season average concentrations of eutrophic parameters (NH4+-N, TP and COD) were higher in the sampling sites of out-of-city area and farmland area, whereas, the higher average concentrations of heavy metals (Pb, Cd and Cr) were concentrated in the out-of-city city area and town area. Pollution of eutrophication was generally worse than the heavy metal in FRS (Fig. 2). In spring, NH4+-N in F6 and COD in F6, 11 were worse than Class V water standard (Table S2); the other parameters higher than Class III standard were: NH4+-N in F7, TP in F7-8, COD in F7 and Cd in F8-9, 11, 14. In summer, NH4+-N in F5, 7-14 were worse than Class III standard; the other parameters higher than Class III standard were: NH4+-N in F2-4, TP in F8, 13, COD in F8 and Cd in F9. In autumn, NH4+-N in F7, 13-14 and TP in F7, 9 were worse than Class III water standard. Overall, the pollution hotspots (>Class III standard) in FRS were mostly (>94%) appeared after the water flowing over the city.

Figure 2 Distributions of average concentration of each water pollution parameter in three seasons in FRS: NH4+-N (A), TP (B), COD (C), Pb (D), Cd (E), Cr (F).

The size of pie chart in each sampling site indicated the average concentration of each pollution parameter of three seasons, and different colors in the pie chart indicated the contributions of different seasons to the average concentration of the individual pollution parameter.

Variation of water pollution among different classified areas

Sampling sites in FRS could be classified into five areas: Headstream, City, Towns, Farmland and Estuary (Fig. 1). There were greatly spatiotemporal changes in each pollution parameter among these five areas (Fig. 3): (1) NH4+-N (average 4.59 ± 0.15 and 0.98 ± 0.04 mg L−1) and TP (average 0.14 ± 0.01 and 0.11 ± 0.02 mg L−1) were all higher in summer and autumn, and increased dramatically from the city area to the estuary; (2) NH4+-N and TP had higher concentrations in city area in spring; (3) concentrations of COD (average 42.07 ± 6.93 mg L−1) were higher in spring with the severe pollution in the city and farmland areas, whereas, COD in summer and autumn was higher in the city area; (4) Cd (average 4.58 ± 0.40 µg L−1) also showed higher concentrations in spring than those in summer and autumn in each area in FRS, however there was no significant temporal variation in Pb and Cr; (5) Pb in spring and summer (16.65 ± 0.85 and 13.07 ± 0.39 µg L−1) was worse than Cd and Cr; and (6) there was no significant spatial variation in Pb after the headstream, whereas, Cd (5.65 ± 1.27, 2.83 ± 0.64 and 2.73 ± 0.32 µg L−1) and Cr (20.00 ± 4.15, 15.45 ± 2.43 and 21.14 ± 3.71 µg L−1) in the town area in three seasons were all higher than other places in FRS.

Figure 3 Distributions of individual water pollution parameters in different areas of FRS in three seasons.

In each graph, solid line indicated Class III surface water quality standard, and dash line indicated Class I surface water quality standard in China (Table S2). Box plots indicated median and first and third quartiles, with whiskers extending to the farthest values within 1.5 times the upper and lower quartiles. Outliers beyond this range were shown as points. The significance of the differences among the median values of sampling areas were indicated by the asterisks (** and *) in the graph at the level p-values of <0.01 and <0.05, respectively. Sample size N ≧ 8.

Effects of different regions of city on the water pollution in FRS

When water flowed out of the city, Middle River had four significantly highest pollution parameters among the three rivers: NH4+-N and Cd in summer, and NH4+-N and Cr in autumn (p were <0.001, <0.001, <0.001, and 0.007), whereas, North River only had two highest parameters (p were <0.001 and 0.003) and there was none in South River (Table 1). Furtherly, Middle River had more than 83% of parameters significantly increased when water flowed through the city, with nearly 5/6 highest increments: NH4+-N (>6 times), TP (>25 times), COD (>7 times), Cr (>3 times) and Cd (>3 times). Contrastingly, North River had none highest increment, and South River had one: Pb (>2 times).

Table 1 Changes of individual water pollution parameter when FRS flowed through the city area.

North River: F2 and 8, Middle River: F3 and 7, South River: F1 and 6 in Fig. 1. SE = standard error of the mean. Superscript lowercase letters and capital letters of mean value indicated the differences were statistical significance at the level p < 0.05 and p < 0.01, respectively.

Summer	NH4+-N
(mg L−1)	TP
(mg L−1)	COD
(mg L−1)	Pb
(μg L−1)	Cd
(μg L−1)	Cr
(μg L−1)	
		mean	S.E.	mean	S.E.	mean	S.E.	mean	S.E.	mean	S.E.	mean	S.E.	
North
river	In
Out	2.28
2.95	0.24
0.27	0.07A
0.31B	0.00
0.08	8.08A
25.38B	2.35 3.00	8.88
9.79	0.81
0.37	1.09A
1.74B	0.11
0.13	7.31a
14.80b	0.88
3.99	
Middle
river	In
Out	1.89A
3.68B	0.23
0.24	0.12A
0.19B	0.01
0.01	5.93A
16.40B	1.21 2.25	9.01
9.84	0.46
0.91	1.40A
2.71B	0.19
0.31	13.67A
23.13B	1.23
5.48	
South river	In
Out	0.28A
0.87B	0.07
0.09	0.03a
0.05b	0.02
0.02	4.01A
11.79B	0.45 2.20	9.77
11.52	0.63
1.42	1.01
1.27	0.08
0.09	10.58a
19.10b	1.76
2.82	
Autumn	NH4+-N
(mg L−1)	TP
(mg L−1)	COD
(mg L−1)	Pb
(μg L−1)	Cd
(μg L−1)	Cr
(μg L−1)	
		mean	S.E.	mean	S.E.	mean	S.E.	mean	S.E.	mean	S.E.	mean	S.E.	
North
River	In
Out	0.29
0.27	0.11
0.04	0.01A
0.07B	0.00
0.01	4.01
6.17	0.69
0.84	6.64
9.78	1.41
1.83	0.67A
1.72B	0.05
0.29	5.65A
12.12B	0.54
1.67	
Middle
River	In
Out	0.24A
1.47B	0.07
0.16	0.01A
0.26B	0.00
0.05	2.04A
15.61B	0.44
3.29	5.89
8.65	0.69
1.17	0.45A
1.54B	0.04
0.14	5.91A
18.74B	0.78
2.28	
South River	In
Out	0.20a
0.33b	0.04
0.03	0.03A
0.20B	0.01
0.08	2.44A
15.91B	0.30
2.88	5.26A
10.84B	1.07
1.50	0.43
1.56	0.05
0.15	7.18
9.40	1.09
1.75	

Relationships among water pollution parameters in FRS

Pearson’s correlation analysis showed there were many strongly positive relationships in the eutrophic parameters and heavy metals (Fig. 4A). Eutrophic parameters: NH4+-N significantly correlated with TP (p =0.0003). Eutrophic parameters and heavy metals: NH4+-N, TP and COD all significantly correlated with Pb (p were 0.0129, 0.0485 and 0.0044), and COD significantly correlated with Cd (p < 0.0001). Principal component analysis (PCA) showed that the first and second principal components (PCs, denoted as Dims in Fig. 4B) explained 42.6% and 33% of the total variance of the heavy metal concentrations in FRS, respectively. Pb and Cd were both positively associated with PC1 with correlations of 66.6% and 79.2%, and Pb was negatively associated with PC2 (correlation: −54.9%); Cr was positively associated with PC2 (correlation: 83%), but also partly associated with PC1 with a correlation of 45.4% (Table S3).

Figure 4 Correlation matrix of eutrophic parameters and heavy metals in the water of FRS (A), and principal components of heavy metals in FRS (B).

In graph (A), the distribution of each variable was shown on the diagonal; the correlation values were shown in the upper triangular portion of the matrix; bivariate scatter plots with fitted lines were displayed in the lower triangular portion of the matrix; statistical significance levels were denoted as “***”, “**”, “*” and “ ⋅ ” corresponding to p-values of <0.001, <0.01, <0.05 and <0.1.

Water quality assessment: changes of WQI with the distance to Baiyangdian

In spring, WQIE mostly exceeded 1.5 in the areas of out-of-city and farmland in FRS (Fig. 5A), which indicated the water was moderately eutrophic based on Class III water standard in China (Table S1). In summer and autumn, WQIE increased gradually from the headstream to the estuary area, which both had a significantly negative linear correlation with the distance to the estuary area (p were < 0.001 and 0.009). WQIE in summer significantly increased from about 0.25 (unpolluted level) to around 2.5 (serious pollution) along FRS with a correlation curve slope of 0.027 (p < 0.001), which were dramatically higher than those in autumn.

Figure 5 Correlation between WQI of eutrophication (WQIE, A) and heavy metals (WQIHM, B) with the distance to the Baiyangdian estuary.

WQI E and WQI HM were calculated based on Class III and I of surface water standards, respectively. Filled points in the two graphs were measured in this study, and unfilled points in the graph (A) were measured by the Environmental Protection Bureau of Baoding. The dashed line was the correlation curve of summer WQI with the distance to the estuary and solid line was the correlation curve of autumn WQI in the two graphs, which were all statistical significance (p < 0.05). The dotted line indicated the mean of all WQI values in each graph, and shaded area indicated the city area.

Based on the Class I water standard, WQIHM were all higher than 2.0 (moderate pollution, Table S1) when FRS flowed over the city in spring, and WQIHM even significantly exceeded 2.5 in the middle areas of FRS (Fig. 5B). WQIHM showed relative lower values in summer and autumn compared with spring, with significantly single peak curve patterns from the headstream to the estuary area (p were 0.002 and 0.021). The curves increased significantly from about 1.0 (unpolluted level) at both ends of FRS to more than 2.0 around the town area (p < 0.001). Based on Class III water standard, WQIHM showed similar spatiotemporal variations along FRS, but all sampling sites were in unpolluted levels (Fig. S1).

Discussion

The eutrophication showed an overall improvement in many China’s inland waters in recent decades (Huang et al., 2019; Zhou et al., 2017). However, the nutrient pollutants were not fully eliminated: the moderate to heavy eutrophication were also found (Guo et al., 2020; Wang et al., 2017; Wu et al., 2017), and the continuous water quality improvement is needed to effectively control the water pollution in their inflow rivers (Lv et al., 2020; Wang et al., 2021a; Wang et al., 2021b; Gao et al., 2021). Our study found a considerable improvement of eutrophication in the inflow river system of Baiyangdian comparing with previous studies, particularly after the establishment of Xiongan New Area in 2017. In the city area of FRS, NH4+-N dramatically decreased from 17.97∼36.92 (2009), 11.34 (2013), 13.33∼27.18 (2014) and 11.89 ± 1.26 (2017) to 1.98 ± 0.28 mg L−1 in our study (Fig. 3); TP decreased even more greatly: from 2.34 (2008), 1.53 (2013), 1.23∼2.15 (2014), 2.25 ± 0.28 (2015) and 2.90 ± 0.18 (2017) to 0.19 ±0.04 mg L−1; whereas, COD (17.65 ± 6.89 mg L−1) moderately decrease comparing with 33.84 ± 4.47 (2005), 31.4 (2013), 54.63 (2014) and 56.93 ± 10.91 mg L−1 (2017) (Dong, 2018; Jia, 2015; Li, 2014; Qiu et al., 2009; Wang et al., 2010). Similar changes of eutrophication were also observed in town and farmland areas (Dong, 2018; Li, 2014; Wang et al., 2010). Eutrophication in the estuary area was not improved so much: NH4+-N from 13.20∼17.27 (2005-2009) to 3.51 ± 0.37 mg L−1, TP from 0.34 (2008) to 0.14 ± 0.01 mg L−1 and COD from 23.13 ± 4.81 (2005) to 13.17 ± 3.20 mg L−1(Qiu et al., 2009; Li et al., 2017; Wang et al., 2010). However, the eutrophic parameters in nearly half of the sample sites in FRS still did not reach the present water quality requirement of Baiyangdian (Class III water standard, Fig. 3), and the NH4+-N and TP in summer and COD in spring in FRS were all higher than those in Baiyangdian (Li et al., 2021b; Meng, Zhang & Shan, 2021; Zhao et al., 2020), which were great threats to the water quality of Baiyangdian (Table S2). In addition, the seriously eutrophic parameters were also observed in many sampling sites, particularly, the NH4+-N and COD in spring and the NH4+-N in summer significantly exceeded the Class V water standard in some hotspots (p < 0.001, Fig. 2), which may have tremendous influences on the water quality of the whole FRS and Baiyangdian. Thus, eutrophication, especially NH4+-N and COD, did not improved in the whole FRS, and the current water quality treatments in Baiyangdian watershed only alleviated the eutrophic pollution in FRS. Furthermore, the eutrophication of Baiyangdian needed to be greatly improved to meet the higher water quality requirement as a natural reserve (Class I water standard, Table S2). All these demonstrated that more precise pollution remediations are needed to deal with the eutrophic pollutants in FRS in the future.

Water pollution caused by heavy metals has caused widespread concern due to their health effects on aquatic animals and humans (Ismail et al., 2016; Rajeshkumar & Li, 2018; Zhang et al., 2017), whereas many previous studies only concerned the eutrophication in the water of FRS and Baiyangdian (Jia, 2015; Li et al., 2021b; Liang et al., 2017; Zhao et al., 2020), and only one research has reported the pollution of heavy metals in the estuary area of FRS: Pb (0.91 µg L−1), Cd (0.08 µg L−1) and Cr (3.75 µg L−1) in the summer of 2016 (Gao et al., 2019). Our study showed that the pollution of heavy metal in FRS was much better than the eutrophic pollution: heavy metals in many sampling sites have nearly reached the Class I water standard (Fig. 3). The average concentrations of Pb, Cd and Cr in FRS stayed in a relative moderate level compared with other aquatic systems globally. In the Dan River drainage, the average concentrations of Cr and Cd were 0.10 and 0.70 µg L−1 (Meng et al., 2016), which were dramatically lower than we found in FRS. Heavy metal pollution in rivers of Greece increased from 1999 to 2019; however, the recent contents of Pb, Cd and Cr were still comparable to the concentrations in FRS (Karaouzas et al., 2021). Whereas, in Houjing River of Taiwan, the average concentrations of Pb, Cd and Cr were 569, 8 and 96 µg L−1 (Vu et al., 2017), in which Pb and Cr were significantly higher than those in FRS. In Huaihe River, the average Pb, Cd and Cr concentrations were 155.60, 69.54 and 22.13 µg L−1 (Wang et al., 2017), and Pb and Cd were significantly higher than Class V water standard and also dramatically higher than what we found in FRS. Furthermore, the similar concentration distributions of heavy metals could indicate the long-distance transportation of heavy metals from the inflow rivers to the downstream lakes (Guo et al., 2020; Meng et al., 2016). Our results displayed the mobility and influence of heavy metals in FRS to the water of Baiyangdian (Fig. 3): the concentrations of heavy metals Cr, Cd and Pb in FRS showed the similar concentration distributions but were all significantly higher than those in water of Baiyangdian (Gao et al., 2019; Meng, Zhang & Shan, 2021; Zhao et al., 2020). The inflow rivers could collect heavy metals from the whole watershed and severely contaminate themselves and downstream lakes (Guo et al., 2020; Meng et al., 2016; Lv et al., 2020). Therefore, in order to completely ameliorate the pollution of heavy metals in Baiyangdian, the sources and routes of heavy metals entering the inflow rivers should be concerned and eliminated.

We did not find any previous study that has shown the spatiotemporal pattern of water pollution in FRS, whereas, the present study displayed dramatically spatial and temporal variations of eutrophication and heavy metal pollution (Figs. 2, 3 and 5), which were consistent with other aquatic ecosystems (Chen et al., 2021b; Guo et al., 2020; Wang et al., 2021a). The captured spatiotemporal patterns would allow us to identify the pollution hotspots and seriously polluted periods in FRS. The eutrophication in FRS showed that NH4+-N and TP increased significantly from the city area to the estuary area in summer and autumn, and NH4+-N and TP in these two seasons were all higher than those in spring (Fig. 3), which could be mainly due to the domestic sewage (Li et al., 2017; Wang et al., 2017) and runoffs from intensive agricultural activities (Tao et al., 2020; Wang et al., 2017), whereas COD contents in spring were higher than those in summer and autumn in FRS, which may be caused by the deteriorated stagnant wastewater and sediment release due to the low flow rate of FRS in spring (Ni, Wang & Wang, 2016; Piao et al., 2010; Zhu et al., 2019). Regarding water quality assessment, WQIE revealed pollution hotspots around the middle area of FRS in spring, and WQIE in summer and autumn linearly increased from unpolluted levels to serious pollution along FRS, showing the pollution hotspots in these two seasons were in farmland and estuary areas (Fig. 5A). The seriously eutrophic water accumulated in the end of FRS could directly enter Baiyangdian and cause contamination, which demonstrated a great influence of inflow rivers’ eutrophic pollutants to the eutrophication in lakes. In contrast to eutrophication, different spatiotemporal variations of the heavy metals were observed in FRS. The contents of Pb were mostly higher than Cd and Cr but without clear spatial variations in FRS (Fig. 3), which indicated Pb could be mainly originated from a constantly line sources along FRS–traffic pollution (Ewen, Anagnostopoulou & Ward, 2009; Vu et al., 2017). Cd and Cr in the town area were all higher than other places in three seasons, which indicated Cd and Cr were likely due to the release from historical polluted sediments in the town area (Peng et al., 2009; Vu et al., 2017) or the industrial wastewater in the town and city areas (Chen et al., 2021a; Li, Chen & Sun, 2021a). The higher concentrations of Cd in spring additionally confirmed that Cd pollution was tremendously influenced by the accumulation effect in stagnant water originating from sediments (Meng et al., 2016; Peng et al., 2009; Wang et al., 2017). WQIHM showed single peak curves along FRS, and increased significantly from unpolluted levels at both ends of FRS to the highest level of pollution just after the city area (around the town area), furtherly demonstrating the pollution hotspots of heavy metals were caused by the sources in the high population density regions (Fig. 5B). Meanwhile, different regions in the city area also greatly affected the water quality of FRS (Table 1): the water flowing through the middle of city was seriously polluted, whereas, the waters flowing through the edges of the city were only slightly polluted. Overall, these spatiotemporal distribution characteristics above clearly clarified the varied pollution hotspots, the seriously polluted periods and the potential sources in FRS. Water quality managements in the future should take serious considerations of these weak parts in current pollution treatments, and formulate and conduct spatiotemporally flexible treatments based on this study to furtherly improve water quality in the Baiyangdian watershed.

Trace elements exhibiting high correlations may share similar analogous behaviors during transformation and migration (Ke et al., 2017; Wang et al., 2017), which can suggest their potential sources and pathways in the water environment (Chen et al., 2021b; Guo et al., 2020; Ke et al., 2017). Pearson’s correlation showed NH4+-N, TP and COD all significantly correlated with Pb (p were 0.0129, 0.0485 and 0.0044) and COD significantly correlated with Cd (p < 0.0001, Fig. 4A), indicating Pb, Cd and eutrophic pollutants were likely originated from similar sources in FRS, such as inflows from domestic sewage and traffic activities (Chen et al., 2021b; Ewen, Anagnostopoulou & Ward, 2009; Guo et al., 2021), runoff from intensive applications of fertilizers and pesticides (Wang et al., 2017; Xia et al., 2018) and release from historical polluted sediments (Ni, Wang & Wang, 2016; Peng et al., 2009; Zhu et al., 2019). PCA showed that Pb was positively associated with PC1 but negatively associated with PC2 (Table S3), which furtherly indicated Pb in FRS could be attributed to the sources of traffic pollution, agricultural practices and historical polluted sediments rather than industrial wastewater discharges (Ewen, Anagnostopoulou & Ward, 2009; Wang et al., 2017). This is consistent with the finding in the spatiotemporal analysis that Pb could be originated from a constantly line sources along FRS (Fig. 3). Cd was also positively associated with PC1, confirming Pb and Cd have similar hydro-chemical characteristics and common sources in the water of FRS (Chen et al., 2021b; Zhuang et al., 2019). In addition, Cd significantly correlated with COD, which could more accurately attribute Cd pollution to the agricultural practices and polluted sediments as COD. In contrast to Pb and Cd, Cr positively associated with PC2 but also partly associated with PC1, which thus can attribute Cr to the sources of sewage releases from industrial activities and polluted sediments (Peng et al., 2009; Vu et al., 2017). All heavy metals partly attributed to the historical polluted sediments were consistent with the results that the WQIHM were relatively higher in the stagnant water in spring (Fig. 5B). Industrial wastewater was imported into Baiyangdian water system since the 1980s, which caused two-thirds of this region to be contaminated and a large amount of pollution has accumulated in the sediments (Zhang et al., 2018; Zhu et al., 2019). Heavy metals released from these sediments could cause a long-term threat to the water quality and aquatic biota health in this region (Ismail et al., 2016; Ke et al., 2017; Rajeshkumar & Li, 2018).

With environmental remediations and increased government financed investments, the water quality (particularly eutrophication) in Chinese inland waters was improved markedly over recent decades (Huang et al., 2019; Zhou et al., 2017; Zhuang et al., 2019). However, in order to completely mitigate the water pollution from Chinese inland waters (include Baiyangdian watershed) in next decades, the importance of controlling the inflow river pollution and understanding its spatiotemporal variations was gradually recognized (He et al., 2020; Tong et al., 2017). We should deeply understand about the pollution sources of inland waters, the main pollutants deteriorating the water quality and the spatiotemporal variations of these pollutants (Huang et al., 2018; He et al., 2020; Wang et al., 2021a). Then, more effective water pollution treatments can be taken with focalizations to reduce the pollution from industrial activities, traffic pollution and agricultural practices, as well as the remediation of polluted sediments. For instance, specific constructed wetlands could be built based on this information for the water quality restoration of the targeted area and pollutants. Therefore, the spatiotemporal changes of the water pollution showed in this study, including the pollution hotspots, serious periods and potential sources, provided important scientific basis for making effective and flexible water quality treatments in the whole watershed of Baiyangdian.

Conclusions

Our study demonstrated a considerable improvement of the eutrophication and a good condition of the heavy metals in the water of FRS after the establishment of Xiongan New Area. However, the eutrophic parameters in nearly half of the sites in FRS still did not reach the present water quality requirement of Baiyangdian, and the heavy metals were mostly associated with traffic pollution, agricultural practices and historical polluted sediments, which were not easily to be controlled and eliminated. In addition, dramatically spatiotemporal changes of pollution in FRS were found in this study, allowing these conclusions: the eutrophication was highest in summer, and the severely eutrophic pollution concentrated around the estuary area, whereas the pollution of heavy metals was relatively similar among three seasons with the prominent pollution around the town area. The serious contamination in these varied pollution hotspots and periods in FRS may have tremendous influences on the water quality of the downstream Baiyangdian. All these findings revealed the weak parts in current pollution treatments, and provided scientific basis for conducting more precise water quality managements to fully eradicate the water pollutants in future.

Supplemental Information

Supplemental Information 1 Water quality classification according to WQI

Click here for additional data file.

Supplemental Information 2 Water quality standards for surface water according to Chinese surface water standards (GB 3838-2002) (unit in μg/L for trace elements, and mg/L for NH4+-N, TP and COD)

Grade I: clean water from headwater and national conservation area that can be used for domestic purposes after simple disinfection, for recreational purposes and irrigation. II: fairly clean water that can be used as domestic water after treatment, for recreational purposes, for fish farming etc., and the area is strictly protected. III: water also can be used for domestic, recreational purposes after suitable treatment. IV: polluted water which can only be used as industrial water after treatment. V: heavily polluted water that should not be used at all.

Click here for additional data file.

Supplemental Information 3 Correlations between variables and dimensions for heavy metals

Click here for additional data file.

Supplemental Information 4 WQI of heavy metals along FRS based on Class I of surface water standard (GB 3838-2002)

Click here for additional data file.

Supplemental Information 5 Raw data of all the measurements in this study

Click here for additional data file.

We are very grateful for some data support provided by Environmental Protection Bureau of Baoding City. The authors would like to thank the editors, the reviewer of Kiran Liversage, and the other two anonymous reviewers for their valuable comments and suggestions on this paper.

Additional Information and Declarations

Competing Interests

Author Contributions

Data Availability

The authors declare there are no competing interests.

Yibing Wang performed the experiments, analyzed the data, prepared figures and/or tables, authored or reviewed drafts of the paper, and approved the final draft.

Yang Wang performed the experiments, analyzed the data, authored or reviewed drafts of the paper, and approved the final draft.

Wenjie Zhang performed the experiments, prepared figures and/or tables, and approved the final draft.

Xu Yao analyzed the data, prepared figures and/or tables, and approved the final draft.

Bo Wang performed the experiments, authored or reviewed drafts of the paper, and approved the final draft.

Zheng Wang conceived and designed the experiments, authored or reviewed drafts of the paper, and approved the final draft.

The following information was supplied regarding data availability:

The raw measurements are available in the Supplementary File.

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
