# Peer review of "Spatiotemporal changes of eutrophication and heavy metal pollution in the inflow river system of Baiyangdian after the establishment of Xiongan New Area"

_PeerJ, doi:10.7717/peerj.13400_

## Round 0.1 · original submission · Minor Revisions

Please revise this manuscript.

Reviewer 1 ·

Basic reporting

Titile. The “water pollution” is too vast, however, the water quality of eutrophication and heavy metals is studied in this manuscript. The title should be clarified and specified.

Section Introductin. The water pollutants, including chemical oxygen demand (COD), ammonia nitrogen (NH4+-N), total phosphorus (TP), plumbum (Pb), cadmium (Cd) and chromium (Cr) should be described and emphasize in Introduction section to clarify the purpose of this research.

Lines220-221. “In spring, NH4+-N in F6 and COD in F6, 11 were worse than Class Ⅴwater standard”.What is the meaning of “worse”? The concentration of WQI parameter may be higher/lower than the WQI guideline/standard.

The pie chart in the Fig.2 is meaningless without the water discharge flux of Fuhe river.

Experimental design

Lines 151-152. “River water were sampled in three seasons: spring (May, headstream was not collected), summer (July) and autumn (October) of 2020, respectively”. Please clarify the reseaons to sample water in the three seasons. Why choose May, July and October? Maybe the “headstream” is not the “head” of Fuhe.

The water transfer from water reservoirs in the Mountain area to Baiyanddian should not be eliminated in the FRS.

Validity of the findings

This research in meaningful for water quality assessment in Baiyangdian lake.

Reviewer 2 ·

Basic reporting

Clear and unambiguous, professional English used throughout.
Literature references, sufficient field background/context provided.
Professional article structure, figures, tables. Raw data shared.
Self-contained with relevant results to hypotheses.

Experimental design

Original primary research within Aims and Scope of the journal.
Research question well defined, relevant & meaningful.
Why only 3 heavy metals, lead, cadmium and chromium are studied, and it is recommended to further review the pollution of other heavy metals
Sampling shall be conducted by the corresponding sampling method of China Standard (GB3838-2002), or by the standard method of EPA

Validity of the findings

There are many literature on the water environment and water ecology of Baiyangdian Lake, and it is suggested to further deepen the review

·

Basic reporting

line 68: change to "due to the measures not taking serious consideration"

line 93: change to "In recent years", and please also clarify the sentence on line 95.

line 107: change to "critically"

line 125: change to "We aimed to"

line 151: change to "River water was"

line 180: change to "due to it being"

line 193: change to "classify". On the next line, please clarify this sentence.

line 198: delete the second "of"

line 243: change to "flowed", also on line 248.

line 283: change to "is needed"

line 294: change "were" to "was"

line 301: change to "threats"

line 309: is the word "processive" here correct?

line 318: change to "showed that the pollution"

line 320: delete "were"

line 323: change to "Greece"

line 329: change to "than what we found in"

line 337: change to "contaminate". On the next line, change to "ameliorate"

line 359: change to "In contrast". Also on line 398.

line 362-363: I do not understand this sentence.

line 367: change "form" to "from"

line 410: change to "recent decades" or "the last decade"

Experimental design

line 342: There were no replicate seasons sampled (i.e. spring, summer and autumn were only sampled during one year), so an experimental design with this limitation cannot be used to understand with consistency any information about seasonal variation. For this reason, I think the word "seasonal" here and else where (e.g. line 345, etc) should be changed to "temporal". I would say that a section of text should be included discussing how there may be seasonal patterns, but until further research has been done with seasonal replication (at least two or three summers, two or three autumns etc) it cannot be known whether the temporal variation seen here is associated with different seasons, or alternatively it may be associated with many other factors that also change across different times.

Validity of the findings

line 198: you should check the assumption of homogeneity of variances for the ANOVAs using Levene's or Cochran's tests. If this assumption is not met, then there may be some transformation of the data required that may change analysis results and affect the validity of the findings.

Additional comments

line 22: it would be good to have some explanation of what this "New Area" is. Is it a newly developed area, or a newly rehabilitated area? etc.

Line 258-261: I do not understand the relevance of correlating different heavy metals and eutrophic parameters with PC1 and PC2. How does this provide useful information? Normally when I do PCA analyses, I use the ordination along the PC axes to correlate different types of variables, e.g. the water quality parameters with samples from different locations, or something like that.

Line 389-392: could you please explain a bit more clearly how exactly the association of Pb with different principal components allowed the conclusion that "Pb in FRS could be attributed to the sources of traffic pollution, agricultural practices and historical polluted sediments rather than industrial wastewater discharges".

Figure 3: it would be good for these graphs to have some indication of whether differences among the sampling areas and/or sampling times are significantly different.

Figure 5: the horizontal dotted line is difficult to see.

---

## Round 0.2 · Minor Revisions

Please revise these minor errors

Reviewer 1 ·

Basic reporting

no comment

Experimental design

no comment

Validity of the findings

no comment

Additional comments

The revision was sufficiently revised according to the reviewers' comments. I think the revision is acceptable for publication.

·

Basic reporting

There were just a few more changes needed to fine-tune the use of English language:

line 206: change to something like: "After the data of pollutant concentrations in three seasons were combined with the GPS positioning of each sampling site, Arc-GIS was used"

line 232: change to "are shown in Fig. 2"

line 357: change to "We did not find any previous study that has shown"

line 411: change to "confirming"

line 447: change to "In addition, dramatic spatiotemporal changes of pollution in FRS were found in this study, allowing the conclusions:"

Experimental design

no comment

Validity of the findings

no comment

---

## Round 0.3 · accepted · Accept

Thank you for responding positively to the comments of the reviewers and for your patience in going through the review process. We appreciate your support of the journal and hope you will publish future work with PeerJ. Congrats